# The selection of a hydrophobic 7-phenylbutyl-7-deazaadenine-modified DNA aptamer with high binding affinity for the Heat Shock Protein 70

Catherine Mulholland[1], Ivana Jestřábová [1,2], Arghya Sett [1], Marek Ondruš[1,2], Veronika Sýkorová[1], C. Lorena Manzanares[1,4], Oliver Šimončík [3], Petr Muller [3] & Michal Hocek [1,2 ✉]

Nucleic acids aptamers often fail to efficiently target some proteins because of the hydrophilic character of the natural nucleotides. Here we present hydrophobic 7-phenylbutyl-7-deaadenine-modified DNA aptamers against the Heat Shock Protein 70 that were selected via PEX and magnetic bead-based SELEX. After 9 rounds of selection, the pool was sequenced and a number of candidates were identified. Following initial screening, two modified aptamers were chemically synthesised in-house and their binding affinity analysed by two methods, bio-layer interferometry and fluorescent-plate-based binding assay. The binding affinities of the modified aptamers were compared with that of their natural counterparts. The resulting modified aptamers bound with higher affinity (low nanomolar range) to the Hsp70 than their natural sequence (>5 μM) and hence have potential for applications and further development towards Hsp70 diagnostics or even therapeutics.

[1] Institute of Organic Chemistry and Biochemistry, Czech Academy of Sciences, Flemingovo nam. 2, CZ-16000Prague 6, Prague, Czech Republic.
[2] Department of Organic Chemistry, Faculty of Science, Charles University in Prague, Hlavova 8, Prague 2, Prague 12843, Czech Republic. [3] Research Centre for Applied Molecular Oncology (RECAMO), Masaryk Memorial Cancer Institute (MMCI), Zluty Kopec 7, 656 53 Brno, Czech Republic. [4]Present address: Department of Chemistry and Center for NanoScience, Ludwig-Maximilians-Universität München, Butenandtstr. 5-13 Haus E, 81377 München, Germany. ✉email: hocek@uochb.cas.cz

Aptamers are short single-stranded nucleic acids that bind to their cognate target with high specificity and affinity[1–3] and can be selected by in vitro systematic evolution of ligands by exponential enrichment (SELEX). Aptamers can be selected against a wide range of possible targets for therapeutic, diagnostic or analytical purposes, spanning from proteins, carbohydrates, enzymes to cells and small molecules. Although having similar binding affinity to antibodies, aptamers offer numerous advantages over them such as they are readily chemically synthesised, non-toxic, have lower immunogenicity and possess higher stability offering a longer shelf life[3]. One of the main disadvantages of generating aptamers is the success rate of SELEX using natural nucleotide libraries, which has been reported to be ~30%[3,4]. This can be attributed to limited available functional groups of the four natural nucleotides and lack of chemical diversity and high hydrophilic nature of the DNA backbone[4–6]. To overcome this drawback, various modified nucleotides have been introduced into the starting SELEX random library generating aptamers with more diverse functional abilities[4,5,7,8]. The concept of using modified aptamers has been around since the 1990s among the first modified aptamers being selected against Thrombin[9,10]. In particular, uracil-modified aptamers have been described extensively against various proteins and other targets[11,12]. There have been numerous reports on using slow off-rate modified aptamers (SOMAmers) by Somalogic, an aptamer-based company focusing primarily on production of modified aptamers with one[13] or two[14] modified bases. In recent years there are further reports on the application of modified aptamers for diagnostics[15] or therapeutics, some of which are in clinical trials for treatment of several diseases[16,17] including macular degeneration[18], acute myeloid leukaemia[19] and coronary heart disease[20]. One of the most recent interesting modified aptamers was the cubamer', a cubane-modified aptamer against the malaria biomarker *Plasmodium vivax* lactate dehydrogenase (PvLDH). This modified aptamer was found to bind within a hydrophobic pocket, offering unique binding characteristics not observed before with any natural aptamer binder[21]. Modified aptamers bearing hydrophobic or protein like groups have been described in the past[7,14,22–25] and are very attractive for developing aptamers with novel functionalities, especially carrying out SELEX against difficult targets or "undruggable" proteins.

Hydrophobic-modified aptamers have been found to increase the binding affinity over their natural counterparts towards respective targets. One example is a modified selection against fibrinogen using a boronic acid modified thymidine reported increased affinity towards its target ($K_d = 3$–30 nM) over its natural sequence ($K_d = 450$ nM)[26]. Another, reports the selection of modified aptamers using random sequence pools of three modified base-appended bases[7]. These authors performed the selections using $U^{ad}$, $U^{gu}$ and $A^{ad}$ to obtain aptamers binding to human β-defensin (HBD-2). Likewise, they discovered that the aptamers obtained from the hydrophobic adenine modified pool bound with the highest affinity towards its target ($K_d = 6.8$ nM).

Over the last 15 years our group has developed various ribo- or 2-'deoxyribonucleotide triphosphates with various chemical functionalities and successfully incorporated these modified dNTPs in polymerase chain reaction (PCR) and primer extension (PEX) using various DNA enzymes (KOD XL, Vent(exo-)) for the synthesis of diverse even hypermodified oligonucleotides[27–29]. Here we designed a new 7-phenylbutyl-7-deaza-2'-deoxyadenosine triphosphate (dA*TP) as a building block containing hydrophobic-modified base for the selection of modified aptamers against the Heat Shock Protein 70 (Hsp70). Hsp70 is a molecular chaperone that is expressed in response to stress and high levels of heat shock proteins, particularly Hsp70 and Hsp90, are found in virtually every type of cancer cells[30–32]. There are many targetable regions and ways to alter Hsp70's chaperone activity, including different druggable sites, *via* allosteric modulation or *via* protein-protein interactions given that chaperones need to bind through exposed hydrophobic regions[33]. Hydrophobic-modiied aptamers can therefore be selected against allosteric modulation or protein-protein interactions. Aside from the therapeutic potential of developing aptamers against Hsp70, there is also a diagnostic potential as antibody-like receptors for in vitro diagnostics[34,35]. Non-modified aptamers against Hsp70 have previously been described[31,36]. Lin et al. described[31] the use of a Tx-01 aptamer, a highly specific aptamer selected against ovarian cancer cells that also interacted with Hsp70. On the other hand Rerole et al. described[36] the selection of peptide aptamers against Hsp70. To the best of our knowledge, there has been no report of hydrophobic-modified DNA aptamers against the Hsp70 protein.

## Results and discussion

**Synthesis of modified nucleoside and dA*TP.** The 7-phenylbutyl-7-deaza-2'-deoxyadenosine was designed as a suitable modification of aptamers because the phenylbutyl moiety contains an aromatic ring and flexible and hydrophobic tether that might allow hydrophobic, π-π stacking or cation-π interactions with the target protein. For the selection and synthesis of aptamers, we needed the modified nucleoside triphosphate (for enzymatic synthesis) as well as the corresponding nucleoside phosphoramidite (for chemical synthesis). The synthesis started with the preparation of 7-iodo-2'-deoxy-7-deazaadenosine (**1**, **dA^I**) by the modified reported procedure[37] where the changes in protocol resulted in overall higher yields (Supplementary Information, section 1.2). **dA^I** was further used in Pd-catalysed Sonogashira cross-coupling with but-3-yn-1-ylbenzene in presence of TPPTS, CuI and TEA which resulted in the alkyne-linked modified nucleoside **2** (**dA^EEPh**, Fig. 1a) in very good yield (93%). Subsequent reduction of the triple bond by catalytic hydrogenation with 10% Pd/C resulted in alkyl-linked modified nucleoside **3** (**dA***, 77%). Final triphosphorylation procedure afforded desired triphosphate **4** (**dA*TP**) in sufficient yield of 19%.

The corresponding protected nucleoside phosphoramidite **7** was synthesised by protection of **3** followed by attachment of the phosphoramidite moiety (Fig. 1b). The first step was the protection of the 5'-OH group of the nucleoside **3** by dimethoxytrityl group providing compound **5** (53%), followed by the protection of the amino group at position 6 by treatment with dimethylformamide dimethylacetal (DMF-DMA) affording nucleoside **6** in 81% yield. Following reaction with 2-cyanoethyl-*N,N*-diisopropylchlorophosphoramidite provided final phosphoramdite **7** in a good yield of 86%. Further details on chemical synthesis can be found in Supplementary Methods, section 1.2.

**The use of modified dA*TP for incorporation into DNA and construction of library.** The synthesised BuPh-modified **dA*TP** was tested as a substrate for KOD XL DNA polymerase in PEX with 31-mer template prb4basII, (Supplementary Fig. S7). KOD XL DNA polymerase was used owing to its reported capability of incorporating modified nucelotides into the library[11,38–40]. The results of the PEX reaction gave full-length product when using the modified **dA*TP** and the rest of natural dNTPs (Supplementary Fig. S7, lane 3) with a slower migration through the gel (caused by the presence of the heavier BuPh-modification), when compared with positive control made up of only all-natural dNTPs (Supplementary Fig. S7, lane 5). These results confirmed that the synthesied **dA*TP** is an acceptable substrate for KOD XL polymerase and therefore may be used to construct the random DNA library for SELEX.

To explore the possible introduction of modified **dA*** into the 65nt library we purchased 5'-biotinylated library (Lib65) (Table 1 and Fig. 2a) containing a random region of 28nt, in-order to

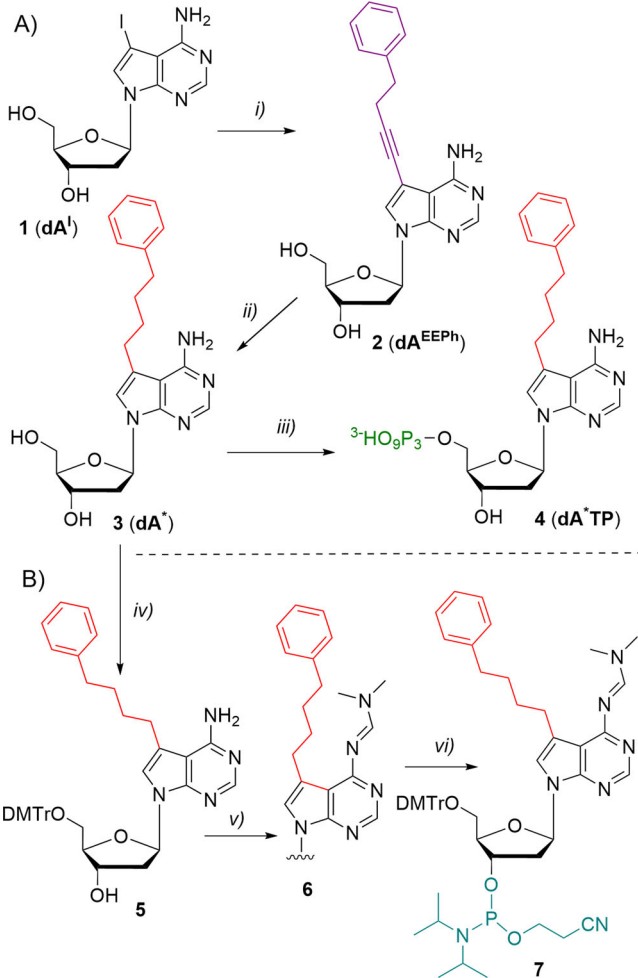

**Fig. 1 Synthesis of PhBu-modified dA\*TP and phosphoramidite 7.**
Reagents and conditions **A**: (i) but-3-yn-1-ylbenzene (10 equiv.), Pd(OAc)$_2$ (0.1 equiv.), CuI (0.1 equiv.), TPPTS (0.1 equiv.), TEA (6 equiv.), MeCN/H$_2$O (1:1), RT, Ar, overnight; (ii) H$_2$ (baloon), 10% Pd/C (0.1 equiv.), MeOH, reflux, overnight; iii) 1. POCl$_3$ (1.2 equiv.), PO(OMe)$_3$, 0 °C, 2 h; 2. (NHBu$_3$)$_2$H$_2$P$_2$O$_7$ (5 equiv.), Bu$_3$N (4 equiv.), DMF (dry), 0 °C, 1 h; 3. 2 M TEAB; Reagents and conditions **B**: iv) DMTrCl (1.2 equiv.), DMAP (0.1 equiv.), pyridine (dry), RT, 6 h; v) *N,N*-dimethylformamide dimethylacetal (14 equiv.), DMF (dry), 40 °C, Ar, 4 h; vi) 2-cyanoethyl-*N,N*-diisopropylchlorophosphoramidite (1.2 equiv.), DIPEA (2.5 equiv.), DCM (dry), 0 °C to RT, 1.5 h.

allow for downstream pulldown and separation of bound from unbound sequences during the selection process. **dA\*TP** again served as a substrate for PEX construction into the DNA library while the reverse primer labelled with 6-FAM at 5′ (P-Rev$^{Lib}$) (Table 1) allowed for visualisation of the resulting extended product by gel analysis. The preliminary PEX test confirmed the successful incorporation of **dA\*** nucleotide into the library (Fig. 2b, lane 3).

We then scaled-up the reaction in order to create enough modified starting material for SELEX. The PAGE gel shows the successful amplification of the 65nt library bearing phenylbutyl-modified **dA\*TP** being readily accepted by KOD XL DNA polymerase and the generation of sufficient quantity (1170 pmol) of modified starting library for the selection (Fig. 2b, lane 4). Figure 2b is also showing that the modified DNA migrated slower through the gel compared to the natural DNA, producing a slightly higher band due to its heavier modification.

**SELEX procedure.** A general outline of the modified SELEX approach developed to select **dA\*TP** modified DNA aptamers against Hsp70 is depicted in Fig. 3. As illustrated, we used a combination of PEX to create the **dA\***-modified library followed by SELEX using magnetic beads based on previous protocols[14,42]. The **dA\***-modified random library was first incubated with pre-washed empty beads containing no target for an initial counter SELEX step. We performed a counter selection directly with the beads before starting round 1 (Table 2) with His-tagged Hsp70 (His-Hsp70), and carried out a further counter SELEX against the beads after every round of positive SELEX. These steps were performed to reduce non-specific aptamer binding to the magnetic particles[43]. ssDNA that became bound to the beads were discarded and the remaining unbound ssDNA was collected in the supernatant and incubated directly with immobilised His-tagged Hsp70 target. Incubation of the target and **dA\***-modified ssDNA was carried out for 30 min at 24 °C, following incubation beads were washed to remove unbound or weakly bound sequences. Sequences that did bind to Hsp70 were amplified using Go-Taq Flexi DNA polymerase, 5′-biotinylated forward primer (P-For$^{Lib}$) and 5′-phosphorylated reverse primer (P-Rev$^{Lib}$) (Table 1) and natural dNTPs (further information provided in the Supplementary Information, gel example provided in Supplementary Fig. S8, lane 2). This enabled us to carry out the remaining SELEX process just as that of a natural SELEX experiment would be carried out. To avoid overamplification and by-product formation, the PCR amplification was monitored after every 3 cycles on a 3% agarose gel and terminated when the band of the correct length of the product was

**Table 1 List of oligonucleotides used in the selection and characterisation of aptamers with binding affinity to Hsp70.**

| Name | Sequence 5′ → 3′ | Length (nt) |
|---|---|---|
| Lib65 [b] | GTGCCAGCTATGCCATTG-28nt-TAGCGTCTATCTCTGCTGC | 65 |
| P-For$^{Lib}$ [a,b,c] | GTGCCAGCTATGCCATTG | 18 |
| P-Rev$^{Lib}$ [a,c,d] | GCAGCAGAGATAGACGCTA | 19 |
| P-For$^{OvH}$ | TCGTCGGCAGCGTCAGATGTGTATAAGAGACAG | 33 |
| P-For$^{OvH+For}$ | TCGTCGGCAGCGTCAGATGTGTATAAGAGACAGGTGCCAGCTATGCCATTG | 52 |
| P-Rev$^{OvH}$ | GTCTCGTGGGCTCGGAGATGTGTATAAGAGACAG | 34 |
| P-Rev$^{OvH+Rev}$ | GTCTCGTGGGCTCGGAGATGTGTATAAGAGACAGGCAGCAGAGATAGACGCTA | 53 |
| P-For$^{Ind}$ | CAAGCAGAAGACGGCATACGAGAT[unique_8nt]GTCTCGTGGGCTCGG | 47 |
| P-Rev$^{Ind}$ | AATGATACGGCGACCACCGAGATCTACAC[unique_8nt]TCGTCGGCAGCGTC | 51 |

[a] natural; [b] 5′-biotinylated; [c] 5′-6-FAM; [d] 5′-phosphorylated.

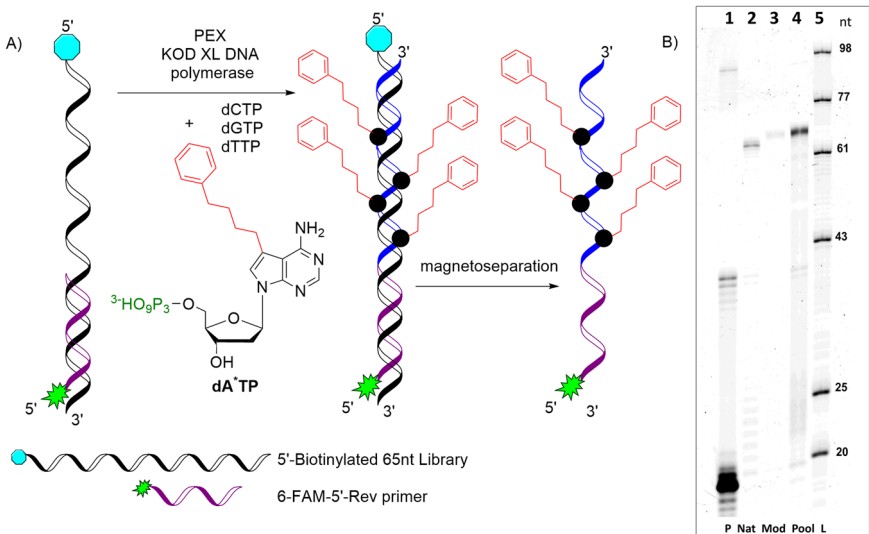

**Fig. 2 Scheme of the PEX construction of modified DNA and denaturing PAGE analysis of PEX reaction with DNA random library using KOD XL DNA polymerase.** Scheme of the PEX (**A**); Denaturing PAGE analysis (**B**): Lane 1 (P): primer, Lane 2 (Nat): DNA library amplified by natural dNTPs, Lane 3 (Mod): DNA library amplified with modified **dA*TP** and natural dCTP, dGTP, dTTP, Lane 4 (Pool): DNA library scaled-up for generation of modified starting library for SELEX, Lane 5 (L): FAM-labelled ssDNA ladder.

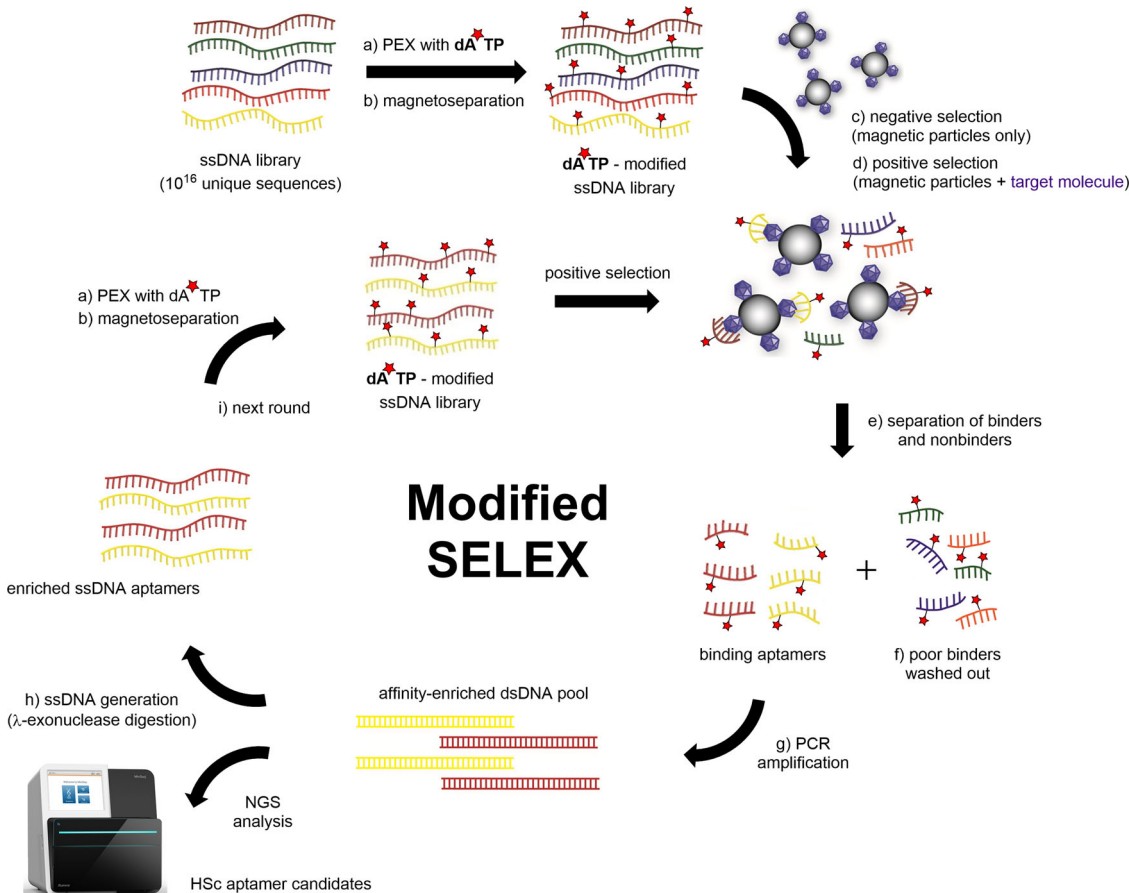

**Fig. 3 Principle of the modified SELEX procedure used in this study (scheme modified from ref. [41]). a** The initial modified library is generated by PEX from 5′-biotinylated ssDNA library followed by **b** single stranded generation via magneto-separation. **c** A counter-selection step against the beads was carried out prior to **d** positive selection with the target. **e, f** The bound ssDNA is separated from the non-binders by washing and **g** the released ssDNA is amplified by PCR using biotinylated forward primer and phosporylated reverse primer, which allows for **h** subsequent ssDNA generation via Lambda exonuclease digestion. **i** The resulting 5′-biotinylated ssDNA is ready for PEX and another round of modified SELEX.

**Table 2 Selection conditions used in each round of the modified SELEX procedure against Hsp70.**

| SELEX round | ssDNA (nM) | Hsp70 (μg) | Time (min) | Competitors (1 mg/mL BSA + 0.1 mg/mL salmon sperm DNA) | Washing conditions |
|---|---|---|---|---|---|
| R1[a] | 500 | 10 | 50 | - | 3 × 1 mL for 1 min |
| R2[a] | 100 | 10 | 50 | - | 3 × 1 mL for 1 min |
| R3[a] | 70 | 10 | 40 | - | 3 × 1 mL for 1 min |
| R4[a] | 70 | 5 | 40 | Yes | 3 × 1 mL for 1 min |
| R5[a] | 70 | 4 | 30 | Yes | 3 × 1 mL for 2 min |
| R6[a] | 50 | 3 | 30 | Yes | 3 × 1 mL for 2 min |
| R7[b] | 15 | 3 | 30 | Yes | 5 × 1 mL for 2 min |
| R8[b] | 15 | 2 | 30 | Yes | 5 × 1 mL for 5 min |
| R9[b] | 15 | 1 | 30 | Yes | 5 × 1 mL for 5 min |

Incubation carried out in 250 μL and at 24 °C. [a] Negative selection performed against the beads only before incubation with the target. [b] Negative selection with His-NPA before incubation with the target.

**Table 3 List of the most abundant aptamer sequences obtained after SELEX against Hsp70.**

| Apt ID | Sequence 5′→ 3′ with primer region underlined | NGS (%) |
|---|---|---|
| HSe-1[a, b] | GCAGCAGAGATAGACGCTATTTCTTA*GA*CCTTTCTA*A*TTTA*A*CTA*CTCA*A*TGGCA*TA*GCTGGCA*C | 8.37 |
| HSe-2[a, b] | GCAGCAGAGATAGACGCTATTCTCTA*GA*CCTTCTA*A*TTTA*CA*TA*CTCCA*A*TGGCA*TA*GCTGGCA*C | 4.57 |
| HSe-3[a, b] | GCAGCAGAGATAGACGCTAA*GGGA*A*GTA*GGA*A*GA*A*GGGA*TGCCCTGCA*A*TGGCA*TA*GCTGGCA*C | 1.58 |
| HSe-4[a, b] | GCAGCAGAGATAGACGCTAGGA*TCTCA*TA*GTTCTTGTA*A*TGA*TA*CTCCA*A*TGGCA*TA*GCTGGCA*C | 1.47 |
| HSe-5[a, b] | GCAGCAGAGATAGACGCTAA*TCGA*TGA*GA*CCTTCTA*A*TTA*A*CTA*TTCCA*A*TGGCA*TA*GCTGGCA*C | 0.78 |
| HSe-6[a, b] | GCAGCAGAGATAGACGCTATCTCTTA*GA*CCTTCTA*A*TTTA*A*CTA*CTCA*A*TGGCA*TA*GCTGGCA*C | 0.74 |
| HSe-7[a, b] | GCAGCAGAGATAGACGCTAGA*TCTCCA*GTCCTTCTA*A*TTTA*A*TA*CTCCA*A*TGGCA*TA*GCTGGCA*C | 0.43 |
| HSe-8[a, b] | GCAGCAGAGATAGACGCTAGTCCTCA*TA*GGCCTTCTA*A*TTA*A*CTA*TTCA*A*TGGCA*TA*GCTGGCA*C | 0.21 |
| HSe-9[a,b] | GCAGCAGAGATAGACGCTAA*GA*CCTCA*TA*GTCCTTCTA*A*TTA*TA*CTCCA*A*TGGCA*TA*GCTGGCA*C | 0.12 |
| HSc-2[c] | GCAGCAGAGATAGACGCTATTCTCTA*GA*CCTTCTA*A*TTTA*CA*TA*CTCCA*A*TGGCA*TA*GCTGGCA*C | |
| HSc-9[c] | GCAGCAGAGATAGACGCTAA*GA*CCTCA*TA*GTCCTTCTA*A*TTA*TA*CTCCA*A*TGGCA*TA*GCTGGCA*C | |
| HSc-9.1[c] | A*GA*CCTCA*TA*GTCCTTCTA*A*TTA*TA*CTCCA*A*TGGCA*TA*GCTGGCA*C | |

[a] Reverse complement sequence purchased 5′-biotinylated; [b] 5′-6-FAM-labelled reverse primer used for PEX production of **dA\*-**modified strand; [c] aptamers synthesised both with and without 5′-6-FAM label; HSe enzymatically synthesised; HSc chemically synthesised.

visually detectable (Supplementary Fig. S8). Next, λ-exonuclease was used for ssDNA generation (Supplementary Fig. S9, lane 3), resulting in 5′-biotinylated ssDNA pool. The 5′-biotinylated ssDNA pool was then used again in a new PEX reaction for amplification of the modified library for the next round of selection Fig. 3. The subsequent selection rounds were carried out in the same manner with decreasing concentrations of Hsp70 and decreasing concentrations of DNA pool (Table 2).

After 6 rounds of SELEX, we introduced a negative SELEX using an alternative His-tagged protein, the protein acidic (PA) N-terminus of RNA-dependent RNA polymerase of Influenza A (His-NPA), in-order to remove aptamers from the pool that were enriched against the His-tag of the Hsp70 target. The incubation time was also decreased between each round and the washing steps increased both in volume and in time to increase the selection pressure.

**Sequencing of aptamers**. After 9 rounds of SELEX we prepared the pools for NGS. Adapter primers (P-For[OvH+For] and P-Rev[OvH+Rev]) (Table 1) were used to amplify each pool and the resulting 150 bp products were purified with AMPure XP magnetic beads according to the manufacturers protocol and analysed by 2% agarose gel electrophoresis using GelRed stain, further information described in Supplementary Information, section 2.6 and shown in Supplementary Fig. S10, lane 2. Index primers (P-For[Ind] and P-Rev[Ind]) (Table 1) were then used to introduce bar codes for the NGS and the resulting 230 bp PCR product was purified with AMPure XP magnetic beads (according to the

manufacturers protocol) and analysed by 2% agarose gel electrophoresis (Supplementary Fig. S10, lane 3). The sequences were imported into AptaSuite software programe for bioinformatic analysis and P-For[Lib] and P-Rev[Lib] fixed primers were used to identify valid sequences (Table 1). Based on frequency in the final round and level of enrichment, nine representative sequences were chosen for initial screening (HSe-1 to HSe-9, Table 3). The random region of each of the aptamers were also aligned using Clustal O (1.2.4) to determine if any motifs exist, the alignment is shown in Table 4.

**Enzymatic synthesis and initial screening of candidates HSe-1 to HSe-9**. In-order for screening of the selected candidates, we first had to generate sufficient quantity of the modified aptamers by PEX as described before in Fig. 2 (further information in Supplementary Information, section 2.7). To do this, we purchased the reverse complement of each aptamer sequence with 5′-biotin (Table 3) and using reverse primer labelled with 5′-6-FAM (P-Rev[Lib], Table 3) along with the relevant dNTPs, the modified strand was extended by KOD XL polymerase. Streptavidin dynabeads were used for pulldown of the antisense strand and NaOH added to release the modified 5′-6-FAM-labelled aptamers which were analysed by PAGE (Supplementary Fig. S11). The results of the PEX reaction showed full-length 65-mer products of each aptamer candidate, with a slower migration through the gel (caused by the presence of the heavier BuPh-modification), than the positive control which contained only natural dNTPs. The gel

**Table 4 Aptamer sequence alignment[a].**

| Apt ID | Random region of the full-length aptamer sequences (5′ → 3′) |
|---|---|
| HSe-1 | TTTCTTA*GA*CCTTTCTA*A*TTTA*A*CTA*CTC |
| HSe-2 | TTCTCTA*GA*CCTTCTA*A*TTTA*CA*TA*CTCC |
| HSe-3 | A*GGGA*A*GTA*GGA*A*GA*A*GGGA*TGCCCTGC |
| HSe-4 | GGA*TCTCA*TA*GTTCTTGTA*A*TGA*TA*CTCC |
| HSe-5 | A*TCGA*TGA*GA*CCTTCTA*A*TTA*A*CTA*TTCC |
| HSe-6 | TCTCTTA*GA*CCTTCTA*A*TTTA*A*CTA*CTC |
| HSe-7 | GA*TCTCCA*GTCCTTCTA*A*TTA*A*TA*CTCC |
| HSe-8 | GTCCTCA*TA*GGCCTTCTA*A*TTA*A*CTA*TTC |
| HSe-9 | A*GA*CCTCA*TA*GTCCTTCTA*A*TTA*TA*CTCC |

[a] The random region of all the selected aptamers was aligned; identical or similar sequences are underlined.

shows successful enzymatic generation of the aptamer candidates and so were assigned as HSe-1 to HSe-9.

To determine the ability of aptamers HSe-1 to HSe-9 to bind to Hsp70, the aptamers were initially screened using a fluorescent-plate-based binding assay (further information supplied in the Supplementary Information, section 2.7), His-tagged Hsp70 was immobilised on a nickel coated 96-well microplate and the 5′-6-FAM-labelled aptamers were incubated with and without protein to determine binding. Fluorescence signal from one replicate of the negative control samples were also included to observe any non-specific binding. Based on the initial screening results in Supplementary Fig. S12, aptamers (HSe-1, HSe-3 to HSe-8) had relatively low fluorescence signals against Hsp70 which suggests poor binding to the target. The results indicate a high number of the most frequent candidates in the round 9 selected pool did not bind to Hsp70 (in particular the most frequent candidate HSe-1), which is in agreement with other SELEX experiments[3,43,44]. Generally, the most frequently observed aptamers do not always imply the best binding candidates. The strongest fluorescence signal was observed for HSe-2 and HSe-9 which suggests that these two aptamers have the strongest binding affinity and therefore, these aptamers were chosen to be chemically synthesised in-house (Table 3) to allow for a larger quantity of the modified aptamers and downstream binding and specificity studies.

**Synthesis and purification of modified dA\*TP aptamers**. In order to have PhBu-modified aptamers in sufficient amount for SELEX, a solid-phase synthesis of the aptamers was applied using corresponding phosphoramidite **7** (Fig. 1). In total, six aptamer candidates were synthesised by chemical solid-phase synthesis with modified phosphoramidite **7** using the 1 μmolar scale. Three different sequences were synthesised and were assigned as HSc-2, HSc-9 and HSc-9.1 (Table 3). Each representative was prepared with or without 6-FAM label at 5′-end resulting in a total of six chemically synthesised aptamers. Further information on oligonucleotide synthesis and NMR spectra can be found in Supplementary Information, section 1.4. The corresponding MALDI spectra of each synthesised aptamer oligo is also shown in Supplementary Figs. S1–S6.

**Binding affinity of HSc-2 and HSc-9**. To characterise the binding affinity of the chemically synthesised modified aptamers HSc-2 and HSc-9, biolayer interferometry (BLI) was used. 50 nM of His-tagged Hsp70 was incubated with various concentrations of HSc-2 and HSc-9 ranging from 0–5000 nM. After BLI screening, the average values were determined using a 2:1 global fitting of target versus ligand and had steady state affinities in the low nanomolar range. The kinetic profiles obtained from the

raw data analysis for both of these aptamers are displayed in Supplementary Figs. S13 and S15. The response data was exported and further analysed by Graph Prism and are displayed in Figs. 4 and 5. The $K_d$ values of HSc-2 and HSc-9, as determined by BLI, was found to be $66 \pm 4$ nM and $36 \pm 2$ nM, respectively. The results showed a typical binding curve, suggesting the aptamer-Hsp70 interaction occurs in a concentration dependant manner (Figs. 4d and 5e). To compare the efficiency of the modified **dA**\*-aptamer with that of its natural counterpart, we tested the binding affinity of the natural sequences of HS-2 and HS-9 (HSNat-2, HSNat-9) against Hsp70 using BLI. The kinetic profiles obtained from the raw data analysis for the natural HSNat-2 and HSNat-9 are displayed in Supplementary Figs. S14 and S16. Again, the response data obtained from these experiments were exported and the data was further analysed by Graph Prism and are displayed in Figs. 4c and 5d. The binding affinity of the natural HSNat-2 and HSNat-9 sequences was found to be >5 μM and $4.65 \pm 0.35$ μM, respectively. In both HSc-2 and HSc-9 modified **dA**\*-aptamers, the binding affinity towards Hsp70 was greatly enhanced compared to that of the unmodified aptamer. In particular, HSc-9 was more than 100-fold higher than that of the natural sequence. Since the nucleotide sequences of the aptamers are identical, the differences in the binding affinities between the natural and modified aptamers suggest that they are attributed to the presence of the hydrophobic modification which apparently plays an important role in the binding of the aptamers to Hsp70. The reason for the enhancement could be due to a change in the folded conformational shape in the presence of hydrophobic **dA**\* nucleotide but also due to some direct interactions of the hydrophobic moieties with the target protein. To further confirm the binding affinities of HSc-2 and HSc-9 obtained from the BLI, we cross-checked the $K_d$ results with a fluorescence plate-based binding assay by incubating His-tagged Hsp70 with various concentrations of 5′-6-FAM-labelled HSc-2 and HSc-9 aptamers in the range of 0–2000 nM. The fluorescence data were exported to Graph Prism and the resulting binding curves are displayed in Figs. 4 and 5. The $K_d$ of HSc-2 and HSc-9, as determined by the plate-based binding assay was found to be $245 \pm 36$ nM and $29 \pm 1.4$ nM, respectively. The $K_d$ values obtained for HSc-2 differed the most between the two binding assays with a 3.7-fold-difference observed between the $K_d$ values. The $K_d$ values obtained for the HSc-9 by both assays were largely in agreement with only a difference of 7 nM. The natural counterpart of each of the HSc-2 and HSc-9 aptamers were also analysed by the plate-based binding assay by incubating His-tagged Hsp70 with various concentrations of 5′-6-FAM-labelled aptamers in the range of 0–10,000 nM. The fluorescence data were exported to Graph Prism and the resulting binding curves are displayed in Figs. 4e and 5g. The results from the fluorescent-based plate assay show no binding of the aptamers to the target with detectable affinity (>10 μM) further demonstrating that the modification in aptamers HSc-2 and HSc-9 must play a vital role in binding. Based on the BLI and the plate-based binding results of **dA**\*-modified and non-modified aptamers, HSc-9, with a binding affinity of $36 \pm 2$ nM by BLI and $29 \pm 1.4$ nM by the plate-based binding assay, was the best candidate and exhibited the highest affinity towards Hsp70 and therefore this aptamer was chosen for truncation and further characterisation studies.

**Truncation of aptamer HSc-9**. Finally, we truncated the HSc-9 aptamer at the 5′ end by removing the fixed primer region (HSc-9.1) (Fig. 5c). This region was chosen as it is the only part of the aptamer where the 2′-deoxyadenosine in the sequence is not replaced by modified **dA**\* nucleotide, even during the SELEX

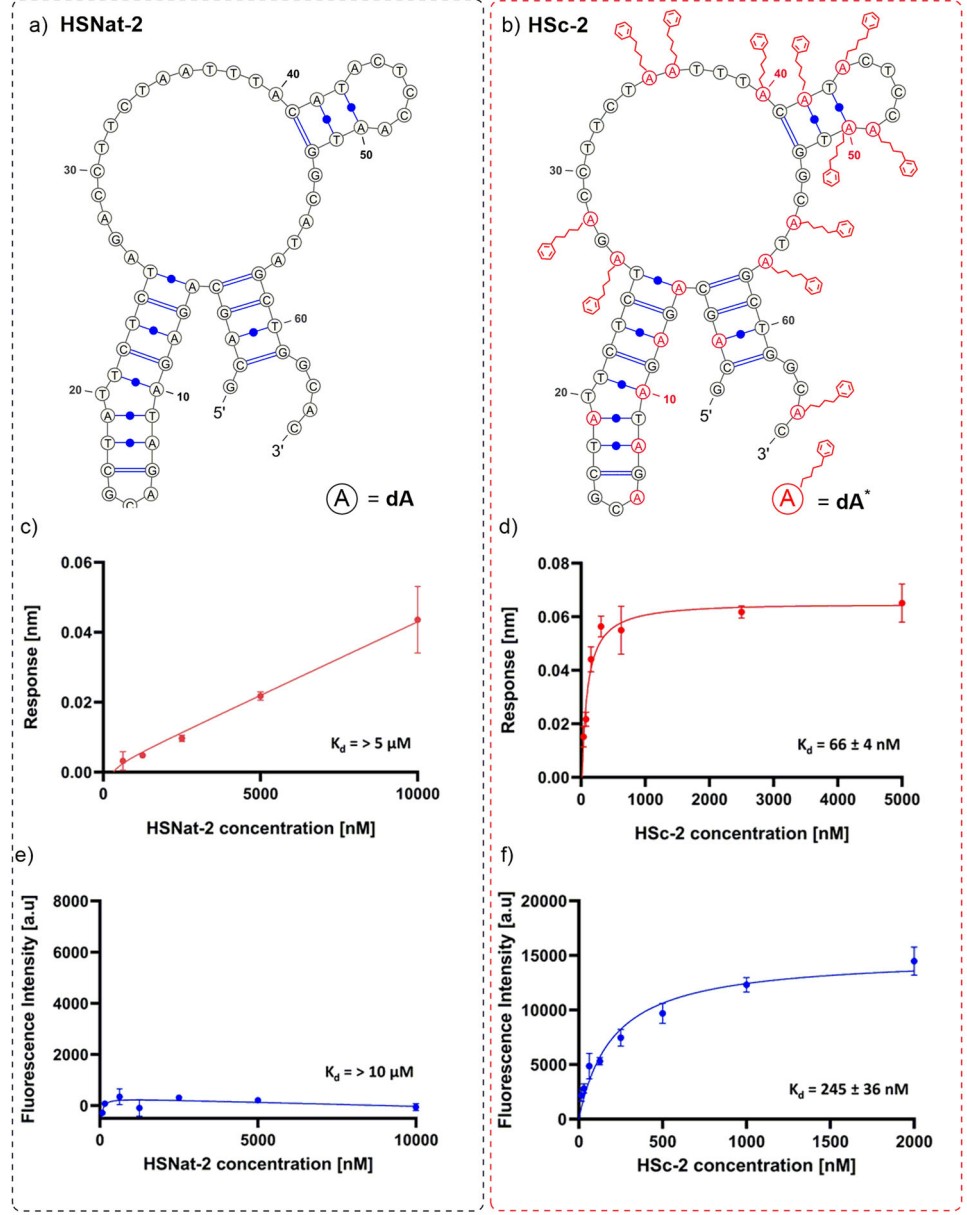

**Fig. 4 Binding affinity results of the natural HSNat-2 sequence and modified HSc-2 aptamer. a, b** the potential predicted secondary structure for the HS-2 aptamers as determined by MFold. BuPh-modifed **dA\*** side chains are highlighted in red for visualisation. **c, d** The corresponding binding curves for HS-2 aptamers as determined by BLI, different aptamer concentrations were used as analyte and the steady state binding affinity [$K_d$] was calculated. **e, f** The corresponding binding curves for HS-2 aptamers as determined by the fluorescent-plate-based binding assay. The raw fluorescent data were exported into Graph Prism and the binding affinity [$K_d$] of each aptamer calculated. Error bars represent the average values of 2 independent experiments.

experiment. We reserved the remainder of the aptamer which contained modified **dA\*** nucleotide, including the fixed 3′ region, to obtain the 46-mer variant HSc-9.1 (Table 3, Fig. 5c). HSc-9.1 was screened by BLI, the average values were determined and the kinetic profiles obtained from the raw data analysis for the truncated HSc-9.1 aptamer is displayed in Supplementary Fig. S17. The response data was exported and further analysed by Graph Prism which is displayed in Fig. 5f. The $K_d$ values of HSc-9.1, as obtained by the BLI showed further enhancement of the binding affinity towards Hsp70 as a result of truncation with a steady state affinity reported of 29.1 ± 2 nM. This binding affinity result was again cross-checked with the plate-based binding assay using synthesised HSc-9.1 that was labelled with 6-FAM fluorophore at the 5′ end. The fluorescence data was exported to Graph Prism and the resulting binding curves are displayed in

Fig. 5i. The plate-based binding assay determined the $K_d$ to be even lower at 14.8 ± 0.3 nM. These results confirm that the truncated aptamer is not dependent on the 5′ fixed sequence, suggesting that this region does not have an important role in target binding. This agrees with other truncation studies[45,46] that have concluded fixed primer sequence regions are less important for aptamer function and may be eliminated.

**Specificity of HSc-9.1 aptamer.** To confirm the specificity of HSc-9.1 toward Hsp70 protein rather than to the His-tag, the binding affinity was evaluated using BLI upon incubation with three alternative His-tagged proteins readily available in our laboratory, namely His-SUMO, the protein acidic C-terminus (His-CPA) and N-terminus (His-NPA) of RNA-dependent RNA

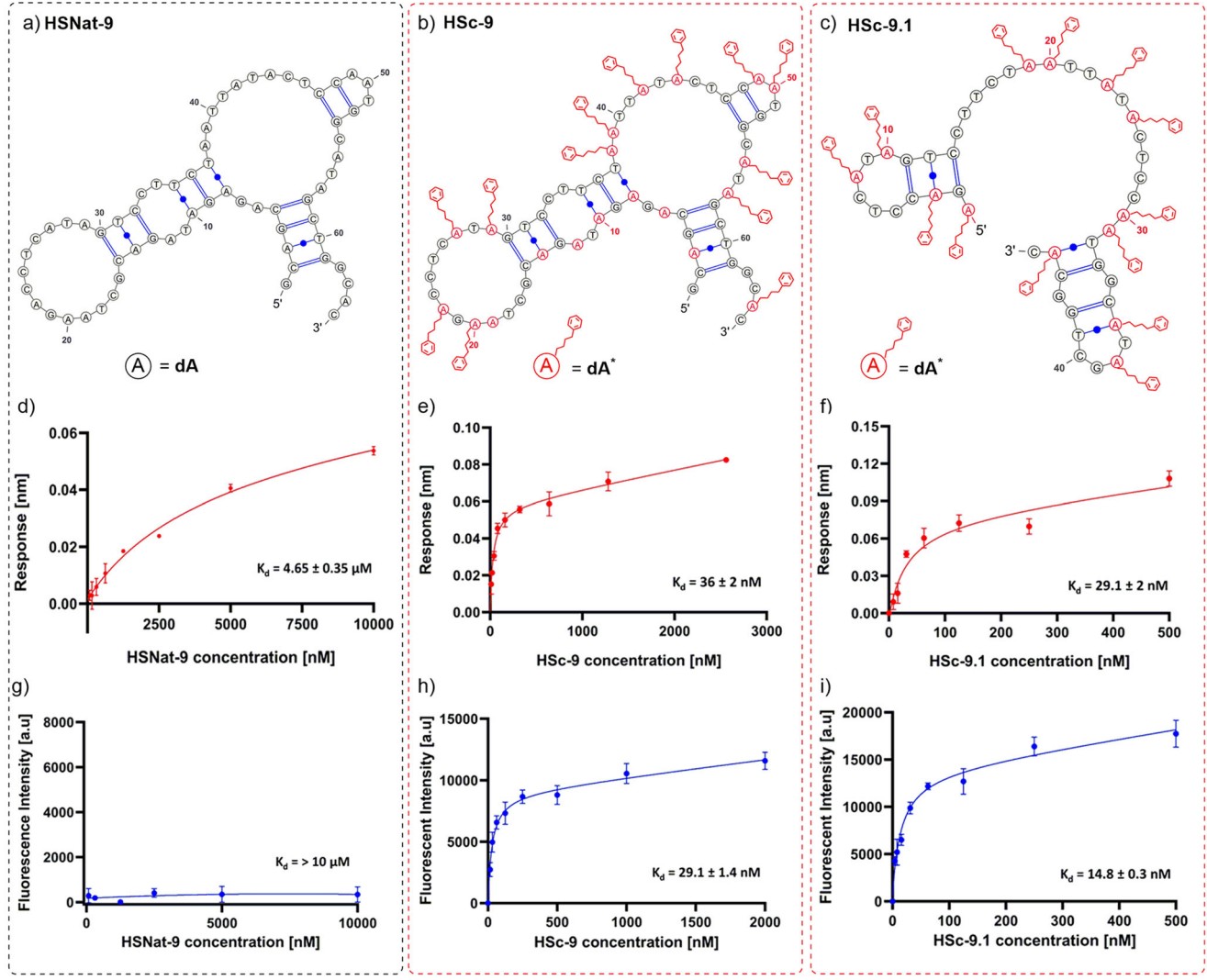

**Fig. 5 Binding affinity results of the natural HSNat-9 sequence, modified HSc-9 aptamer and the modified truncated HSc-9.1 aptamer. a–c** The potential predicted secondary structures for the HS-9 aptamers as determined by MFold. PhBu-modifed **dA\*** bases with side chains are highlighted in red for visualisation. **d–f** The corresponding binding curves for HS-9 aptamers as determined by BLI, different aptamer concentrations were used as analyte and the steady state binding affinity [$K_d$] was calculated. **g–i** The corresponding binding curves for HS-9 aptamers as determined by the fluorescent-plate-based binding assay. The raw fluorescent data were exported into Graph Prism and the binding affinity [$K_d$] of each aptamer calulated. Error bars represent the average values of two independent experiments.

| Table 5 The binding affinity of HSc-9.1 against different proteins. | | |
|---|---|---|
| **Target Protein** | **BLI [$K_d$]** | **Plate assay [$K_d$]** |
| His-Hsp70 | 29.1 ± 2 nM | 14.8 ± 0.3 nM |
| His-NPA | 1550 ± 78 nM | 1250 ± 70 nM |
| His-CPA | 1241 ± 56 nM | 1001 ± 65 nM |
| His-SUMO | 1318 ± 85 nM | 1110 ± 50 nM |
| SBP-Hsp70 | 28.5 ± 1.2 nM | - |

polymerase of Influenza A (Supplementary Fig. S20a) and we also cross-checked these $K_d$ results with a fluorescence plate-based binding assay by incubating the alternative His-tagged proteins with various concentrations of 5′-6-FAM-labelled HSc-9.1 (Supplementary Fig. S20b). The $K_d$ values of the HSc-9.1 towards the alternative proteins were all determined to be greater than 1 μM (Table 5) as compared to double-digit nanomilar affinity toward His-Hsp70.

To further confirm that aptamer HSc-9.1 can bind to Hsp70 even in the absence of the His-, we tested this aptamer against Hsp70 containing an alternative tag, namely, a streptavidin binding peptide (SBP) tag. We screened by BLI with the response data being exported and analysed by Graph Prism and is displayed in Supplementary Fig. S21. The $K_d$ values of HSc-9.1 towards SBP-tagged Hsp70 was reported as 28.46 ± 1.2 nM. This is largely in agreement with the binding affinity obtained by the His-tagged Hsp70 ($K_d$ 29.1 ± 2 nM). These results confirm that HSc-9.1 aptamer is binding directly with the Hsp70 and that neither of the tags play an important role in aptamer binding.

In order to test the sequence specificity of the HSc-9 aptamer, we synthesised three different scrambled ONs (Scr1-3) containing the same numbers of each nucleotide and modifications but in a randomly different order. The binding affinity was determined by BLI and the response data are displayed in Supplementary Fig. S22a. We cross-checked these $K_d$ results with a fluorescence plate-based binding assay by incubating the His-tagged Hsp70 with various concentrations of 5′-6-FAM-labelled HSc-9

**Table 6 Binding affinities of aptamers and their scrambled alternatives with His-Hsp70.**

| Aptamers | BLI [$K_d$] | Plate assay [$K_d$] |
|---|---|---|
| HSc-2 | 66 ± 4 nM | 245 ± 6 nM |
| HSNat-2 | >5 µM | >10 µM |
| HSc-9 | 36 ± 2 nM | 29.1 ± 1.4 nM |
| HSNat-9 | 4.7 ± 0.4 µM | >10 µM |
| HSc-9.1 | 29.1 ± 2 nM | 14.8 ± 0.3 nM |
| Scr1 | 560 ± 15 nM | 536 ± 35 nM |
| Scr2 | 620 ± 55 nM | 687 ± 43 nM |
| Scr3 | 700 ± 37 nM | 775 ± 21 nM |

Supplementary Fig. S22b. The $K_d$ values of the scrambled HSc-9 towards the Hsp70 protein were reported to be at least one order of magnitude higher than the parent aptamer sequence (Table 6) which confirms that the sequence (rather than just the presence of the modifications) is crucial for the high affinity and specificity.

**HSc-9.1 as a capture aptamer for Hsp70.** Finally, to further confirm that HSc-9.1 is binding to Hsp70 and that the aptamer could function in a different orientation, we conducted a sandwich ELISA to investigate the capture ability of HSc-9.1 when immobilised (further information supplied in Supplementary Information, section 2.14). HSc-9.1 was biotinylated at the 3′-end via Biotin-16-dUTP using terminal deoxynucleotidyl transferase (TdT), so that it could be immobilised to streptavidin plates. We used Alexa-fluor anti-Hsp70 antibody as a detection agent and first performed a series of controls to validate the assay (Supplementary Table S2 and Supplementary Fig. S18). The results show that when immobilised HSc-9.1 was incubated with Hsp70 and antibody against Hsp70, increased fluorescence was observed. When Hsp70 was not added and the capture aptamer incubated with antibody only and binding buffer only, the fluorescence signal remained similar to that of the negative binding buffer control. Similar low fluorescence results were observed when Hsp70 was added directly to the plate followed by antibody, indicating that Hsp70 was not binding non-specifically to the plate. The results demonstrate that the reason for the high fluorescence in control 1 was that HSc-9.1 was indeed capturing Hsp70 target and therefore there is binding between antibody and Hsp70. These results confirmed that the removal of the aptamer in various control tests prevents binding of Hsp70 providing additional evidence that the aptamer is specifically interacting with Hsp70. HSc-9.1 was then used for the capture of various concentrations of Hsp70 using the same ELISA conditions described above. Supplementary Fig. S19 shows a linear increase in fluorescence when the concentration of Hsp70 was increased with a wide detection range confirming that HSc-9.1 captures Hsp70 in a concentration dependant manner.

## Conclusions

Hydrophobic-modified aptamers have been found to increase the binding affinity over their natural counterparts[7,21,24]. Herein, we report the synthesis of a hydrophobic PhBu-modified **dA\*TP** and its use in the selection of chemically modified **dA\***-aptamers with high affinity against Hsp70. Although Hsp70 aptamers have been selected in the past, this article describes the generation of hydrophobic-modified aptamers against Hsp70. To begin with, we performed nine rounds of selection against Hsp70 using the PhBu-modified **dA\*TP**. The pool was sequenced and 9 aptamers were selected based on the overall frequency in the last round. Initially for screening of the candidates, the aptamers were enzymatically synthesised via PEX reaction using KOD XL DNA

polymerase and reverse primer that was 5′-6-FAM-labelled. Following single-stranded generation, the 5′-6-FAM-labelled aptamers (HSe-1 to HSe-9), were incubated with Hsp70 pre-immobilised onto black Nickel plates. The fluorescence intensity was measured and from the initial screening results aptamers 2 and 9 demonstrated the highest binding towards Hsp70 and therefore these sequences were chosen to be chemically synthesised in-house. Next, we determined the binding affinity of the modified aptamers and compared it with their natural counterpart using two methods, BLI and a fluorescent-plate-based binding assay. Based on BLI and plate-based binding assay results we produced modified aptamers that can bind with high affinity and specificity to Hsp70. The results of natural versus modified aptamers differed significantly in binding affinity towards Hsp70 with modified aptamers having higher binding affinity towards the target (by 200 fold) and are in agreement with previous studies carried out between modified aptamers and their natural counterparts[7,10,25]. Importantly, the modified aptamers described here were able to bind to Hsp70 with far greater affinity than their unmodified sequences demonstrating that the modification in the sequence plays an important and useful role in binding. In summary, we succesfully generated high-affinity aptamers with a hydrophobic base modification specific for Hsp70 and demonstrated that these aptamers bind with higher affinity than their natural counterpart. With further development, these modified aptamers have significant potential for use in therapeutic or diagnostic applications.

## Methods

Complete experimental part including the synthesis of the modified nucleoside, **dA\*TP** and phosphoramidite, the NMR and MALDI data, detailed procedures on enzymatic synthesis, gel analysis of aptamers and further methods can be found in the Supplementary Information. Only the major procedures are given below:

**The use of modified dA\*TP for incorporation into DNA and construction of library.** The 65-nucleotide combinatorial DNA library with constant forward and reverse regions and a central randomised sequence of 28 nucleotides and biotinylated at the 5′ end (5′-biotin-GTGCCAGCTATGCCATTG-N$_{28}$-TAGCGTC-TATCTCTGCTGC-3′) Table 1, was synthesised and HPLC purified by TriLink (USA). The ssDNA library and oligonucleotides were dissolved in nuclease-free water from VWR (USA) to a final concentration of 100 µM. A 6-carboxyfluorescin (6-FAM) labelled reverse primer (P-Rev$^{Lib}$) was used in a PEX reaction to synthesise labelled modified **dA\*TP** antisense-DNA library (Fig. 2). The PEX reaction mixture with a total volume of 20 µL for a single reaction contained: 5′-biotinylated DNA library (100 µM, 2 µL), 5′-6-FAM-labelled reverse primer (P-Rev$^{Lib}$) (100 µM, 2.4 µL), natural dCTP, dGTP, dTTP (5 mM, 2 µL), **dA\*TP** (3.8 mM, 2.63 µL), KOD XL DNA polymerase (2.5 U/µL, 1 µL) and KOD XL reaction buffer (10×, 2 µL). The PEX reaction mixture was incubated for 5 minutes at 95 °C, followed by 1.5 minutes at 55 °C, followed by 2 hours at 60 °C. The reaction was terminated with the addition of PAGE stop solution (10 µL) and denatured for 3 minutes at 95 °C. Samples were analysed by PAGE and visualised using fluorescence imaging (Supplementary Fig. S1). Dynabeads MyOne Streptavidin C1 (10 mg/ml) were used to capture the 5′-biotinylated dsPEX library. The beads were vortexed thoroughly and 100 µL (1 mg) of the mix was transferred to a new tube. For partitioning of the beads, a magnetic stand was used to capture the beads and the supernatant was removed. The beads were first washed with 1 mL of 1× binding and wash buffer (B&W) (5 mM Tris, 0.5 mM EDTA, 1 M NaCl, pH 7.5). The PEX reaction was diluted to 100 µL with nuclease-free water and added to the beads containing 100 µL of 2× B&W buffer for a total volume of 200 µL. The reaction was incubated for 1 hour at 24 °C with gentle end to end rotation. The beads were captured by magnet and washed 3 times with 1 mL of 1× B&W buffer followed by 2 times with 1 mL of nuclease-free water. For the release of the single stranded modified DNA library, the washed bead pellet was resuspended thoroughly in 100 µL of 50 mM NaOH solution for 10 minutes. The magnetic beads were separated and the supernatant was transferred to a clean Eppendorf vial containing 10 µL of 10× binding buffer (250 mM HEPES, 50 mM MgCl$_2$, 3 M KCl, pH 7.5). The sample was then neutralised by adding 10 µL of 0.5 M HCl.

**Immobilisation of His-Hsp70 to magnetic beads.** His-tagged Hsp70 was immobilised to HISPUR-Ni-NTA magnetic beads to perform the selection of aptamers. The beads were vortexed thoroughly and 10 µL (0.1 mg) of the mix was transferred to a new tube. For partitioning of the beads, a magnetic stand was used to capture the beads and the supernatant was removed. The beads were washed 3

times with 500 μL of 1× wash buffer (PBS, 0.05% Tween 20). A three-fold excess of His-tagged Hsp70 (30 μg) was added to the magnetic beads in-order to allow protein saturation to the beads, the reaction was incubated for 1 hour at 24 °C with gentle end to end rotation. The beads were captured by magnet and washed three times with 500 μL of wash buffer followed by 2 times with 500 μL of PBS only and then eluted in original starting volume of 10 μL with PBS.

**SELEX procedure**. In total, nine SELEX rounds were performed (Table 2). Before incubation with Hsp70, the modified ssDNA library containing 500 pmol was denatured by heating at 90 °C for 5 min in 250 μL of 1× binding buffer (25 mM HEPES, 5 mM MgCl$_2$, 300 mM KCl, pH 7.5) and cooled at room temperature for 30 min. In-order to reduce the non-specific interactions of the pool with the surface of the beads, a negative selection step against 10 μL of the clean HISPUR-Ni-NTA magnetic beads without protein was performed by incubating the folded modified DNA library with the beads for 45 min at 24 °C with gentle end to end rotation. The non-specific ssDNA that bound on the beads, was separated by magnet and the supernatant which contains the remaining ssDNA-modified pool, was used directly for the first round of SELEX. The remaining modified ssDNA, was mixed and incubated with 10 μg of immobilised Hsp70 on beads in a final reaction volume of 250 μL in 1× binding buffer. Incubation was carried out at 24 °C for 1 hour with gentle end-to-end rotation. After the binding reaction, the beads were separated by magnet and the supernatant containing the unbound sequences was discarded. The pellet was washed 3 times with 1 mL of wash buffer (25 mM HEPES, 5 mM MgCl$_2$, 300 mM KCl, pH 7.5 + 1 mg/ml of BSA, 0.1 mg/ml salmon sperm DNA and 0.1% Tween 20) to facilitate the removal of non-binding sequences. Next, 200 μL of nuclease-free water was added to the beads and the sample was heated to 95 °C for 5 min to release the ssDNA. The beads were separated and the supernatant was quickly collected and transferred to a clean 1.5 mL Eppendorf DNA LoBind vial. PCR amplification of the protein-bound modified ssDNA pool was performed using 5′-biotinylated forward primer (P-For$^{Lib}$) and 5′- phosphorylated reverse primer (P-Rev$^{Lib}$) (Table 1) and natural dNTPs. The PCR was performed in a final reaction volume of 50 μL which containing P-For$^{Lib}$ and P-Rev$^{Lib}$ (500 nM, 2.5 μL each), MgCl$_2$ (5 mM, 2.5 μL), natural dATP, dCTP, dGTP, dTTP (0.3 mM, 0.6 μL), Go-Taq Flexi DNA polymerase (1.25 U, 0.25 μL), 5× colourless Go-Taq Flexi Buffer (1×, 10 μL) and 10 μL of ssDNA pool. The reaction was performed under the following cycling conditions: 95 °C for 2 min; followed by cycles of 95 °C for 15 s, 68 °C for 30 s, 72 °C for 30 s and a final extension of 72 °C for 2 min. The visualisation of PCR products was done by the intercalating agent GelRed as described in the Supplementary Information, section 2.1. Once the correct size of product appeared on the gel the amplification was terminated and the double-stranded DNA products were pooled and purified using NucleoSpin gel and PCR clean up system (Machery and Nagal). For separation of the PCR product and recovery of ssDNA for subsequent rounds, Lambda exonuclease digestion was used. The reaction mixture with a total volume of 50 μL for a single reaction contained Lambda exonuclease (10 U, 1 μL), Lambda exonuclease reaction buffer (10×, 5 μL) and 45 μL of purified dsDNA. The reaction mixture was incubated at 37 °C for 60 minutes and analysed by 3% agarose gel electrophoresis and visualised by GelRed as previously described in Supplementary Methods, section 2.1. 5′-biotinylated ssDNA products were then purified using NTC Buffer and the NucleoSpin gel and PCR clean up columns according to manufacturer's instructions and eluted in 50 μL of nuclease-free water.

**Binding affinity of aptamers using BLI**. Biolayer interferometry (BLI) was used for the determination of binding affinity [K$_d$]. A nickel-coated biosensor was used to capture His-tagged Hsp70. Prior to each BLI assay, the sensors were pre-hydrated in 200 μL of 1× binding buffer (HEPES, 5 mM MgCl$_2$, 300 mM KCl, pH 7.4) for at least 10 min. All solution volumes were 200 μL and all steps were prepared in 1× binding buffer. All candidates, modified HSc-2 and HSc-9 and natural HSNat-2 and HSNat-9, were diluted in 1× binding buffer at various concentrations (156, 312, 625, 1250, 2500, and 5000 nM, 200 μL each) and the samples were heated to 95 °C for 5 min and cooled to 24 °C for 30 min to allow for aptamer folding. In the case for the truncated HSc-9.1 aptamer, it was diluted in 1× binding buffer at lower concentrations (7.8, 15.6, 31.3, 62.5, 125, 250, 500 nM, 200 μL each). The experiment protocol consisted of five steps: (1) baseline for 60 s, (2) loading for 300 s, (3) baseline 2 for 30 s, (4) association for 300 s, and (5) dissociation for 300 s. The baseline solution was 1× binding buffer, the loading solution contained His-tagged Hsp70 (50 nM, 200 μL) in 1× binding buffer, the association solution contained varying concentrations of aptamers HSc-2, HSc-9, and natural HSNat-2 and HSNat-9 (156, 312, 625, 1250, 2500 and 5000 nM, 200 μL each) and the dissociation step took place in 1× binding buffer (200 μL). The response data obtained were double-reference subtracted from the blank controls, which contained (1) binding buffer only and (2) ligand loading and binding buffer only. The response data obtained from the experiment were analysed and exported from the Octet Data Analysis Software 8.2.0.7.

**Binding affinity of aptamers using a fluorescent-plate-based binding assay**. The binding affinity of aptamers HSc-2 and HSc-9 was cross-referenced using the fluorescent-plate-based binding assay as described in Supplementary Information, section 2.7 for screening of the aptamers. The assay was performed in

black nickel-coated 96-well plates as follows: wells were washed three times with 200 μL of wash buffer (HEPES, 5 mM MgCl$_2$, 300 mM KCl, 1% BSA and 0.05% Tween 20). 100 μg of His-tagged Hsp70 in 100 μL of 1× binding buffer (25 mM HEPES, 5 mM MgCl$_2$, 300 mM KCl) was incubated in the wells for 1 hour at 24 °C with shaking at 300 rpm. As a negative control, wells containing 1× binding buffer only and without Hsp70 were included for background subtraction. All wells were washed with wash buffer and blocked with 200 μL of 3% BSA for 1 hour at 24 °C with shaking at 300 rpm. 5′-6-FAM-labelled HSc-2 and Hsc-9 aptamer candidates were serially diluted from 2000 nM to 15.6 nM and 100 μL of each aptamer was heated to 95 °C, cooled for 30 minutes at room temperature to allow folding, and added to wells containing Hsp70 and also control wells without Hsp70 for 1 hour at 24 °C with shaking at 300 rpm. As a positive control, 100 μL containing 5 μg of anti-Hsp70 antibody (labelled with Alexa-fluor) was added to wells with and without Hsp70 and incubated for 1 hour in the dark at 24 °C with shaking at 300 rpm. Wells were then washed five times with 200 μL of wash buffer and finally 100 μL of 1× binding buffer was added before the measurement. Fluorescence intensity was measured using Tecan microplate reader as described in the Supplementary Information, section 2.1.

**Specificity testing of truncated aptamer HSc-9.1**. The specificity of the truncated aptamer HSc-9.1 was tested against other His-tagged proteins readily available in our laboratory. The specificity testing was carried out by the plate-based binding assay as described above and in Supplementary Information, section 2.7. 100 μg of each His-tagged protein namely, His-SUMO, the protein acidic (PA) C-terminus and N-terminus of RNA-dependent RNA polymerase of Influenza A (His-CPA, His-NPA), in 100 μL of 1× binding buffer (25 mM HEPES, 5 mM MgCl$_2$, 300 mM KCl) was incubated in the wells for 1 hour at 24 °C with shaking at 300 rpm. As a negative control, wells containing 1× binding buffer only without each of the His-tagged proteins were included for background subtraction. Fluorescence intensity was measured using Tecan microplate reader as described in previously.

## Data availability

Data are given in Supplementary Information. Additional primary data including raw data from ELISA and BLI and NGS sequencing are available from the repository: https://doi.org/10.48700/datst.cjqbt-kn491. NMR spectra are available in Supplementary Data 1.

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

## Acknowledgements

The work was mainly funded by Czech Science Foundation (20-00885X to C.M., I.J., A.S. and M.H.). Additional support to co-authors by European Regional Development Fund; OP RDE (No. CZ.02.1.01/0.0/0.0/16_019/0000729 to V.S.), Czech Science Foundation (22-17102S to P.M. and O.S.), and the Ministry of Health Development of Research Organisation, MH CZ - DRO (MMCI, 00209805 to P.M. and O.S.) is also acknowledged.

## Author contributions

C.M. and M.H. designed the study, analysed results and wrote the paper. C.M. performed the biochemistry, selection experiments and interaction studies with important contribution of A.S., L.M., M.O., and V.S. I.J. synthesised all modified nucleosides, nucleotides and oligonucleotides. O.S. and P.M. expressed and purified the His-tagged Hsp70.

## Competing interests

The authors declare no competing interests.
