## [Peer Review File · Communications Chemistry]

Reviewers' comments:

Reviewer #1 (Remarks to the Author):

The study by Hocek and co-workers describes the identification of DNA aptamers with hydrophobically modified adenine residues binding to Hsp70. The study describes the synthesis of the modified ATP and phosphoramidite, the use of these to generate modified DNA libraries and their application to SELEX for enriching the respective aptamers. The enriched libraries were analysed regarding sequence content and the interaction properties of one representative aptamer are shown in the main part of the manuscript, including truncation and proof of interaction-dependency on the hydrophobic modification. The study is very well written and described in a scholar manner, describes a modified aptamer binding to Hsp70 and might be acceptable for publication, provided the comments below are considered:

1. references of relevant studies in the fields are given but this reviewer missed the work by Hirao on hydrophobic base-pairs and Mayer on click-modified (hydrophobic) modified aptamers/clickmers, which have proven useful as potential therapeutics and diagnostics.
2. Abstract: Typo Aptam,ers needs to be aptamers
3. Controls: the study shows data using a naive DNA aptamer (which revealed fairly high affinity to Hsp70) and the corresponding (selected) modified aptamer with much higher affinity. However, missing here data on scrambled versions of the modified aptamer, having the same amount of modifications and of each other nt, but in a different order. The Aptamer Society recently published a white paper on minimal criteria necessary to meet prior to publish new aptamer/target pairs, and such controls are identified as essential.
4. Along the same lines as 3: Which modifications at which positions are important for maintaining binding? Are all 13 modifications necessary or can be some of them omitted? This would be valuable data for further studies on the aptamer, but might go beyond the scope of this ms.

Reviewer #2 (Remarks to the Author):

In this manuscript the authors present the selection of hydrophobic modified DNA aptamers against HSP70. The paper incorporates rigorous chemistry of the synthesis of a modified dATP which has a hydrophobic sidechain. This was used in a SELEX to discover hydrophobic modified aptamers, then aptamers characterised by various biophysical methods. The chemistry of the paper is very strong, overall I would consider this could be suitable for Comms Chem - ideally this might include a structure of the complex as well, but the extent of characterisation is also very good, although not quite exceptional without X-ray structure.

I have these particular comments to address:

1. During selection it seems only magnetic particles instead of a different protein+magnetic particle was used as counterselection. Therefore, one must be careful with controls that this is not a general protein binder, particularly when considering the hydrophobic modification. BLI data (eg. S13) should be shown with an alternate protein to HSP70 as a control. In both directions (ie. also S14). Also figures 4 and 5 - these need more controls of alternative proteins and alternative DNA sequences overlaid.
2. I would advise switching some data of binding to main figures. The raw data of S13 and S15 should be the main figure with the graph prism export figure 4 and 5 being sub-figures within that to allow the reader to see quality of raw data without referring supplementary.

3. Specificity is shown in Figure 6 but this is more qualitative at a single concentration instead of quantitative. Ideally this should actually provide a K_d to each of the targets (or none shown if cannot be determined) through a range of concentrations.

4. Is there data the aptamer can bind when the his tag is removed from Hsp70?

5. A table showing relative affinities of different sequences with and without modification would be helpful - it is difficult to dig this out of the text describing Fig 4 and Fig 5 on page 15. Also the comparison of BLI and fluorescent plate data.

6. For Table 3 is there some sort of alignment? Are these sequences related?

7. A little more discussion regarding biological relevance of aptamers to detect Hsp70 would be helpful.

In summary this is quite a strong paper particularly from the chemistry perspective. In my view after revisions it may be suitable for publication.

Reviewer #3 (Remarks to the Author):

The use of nucleic acid libraries with modifications that increase chemical diversity has substantially broadened the scope of targets for which high-affinity aptamers can be identified. Mulholland et al. report the selection of DNA-based aptamers with a new hydrophobic aromatic modification introduced at the 7-position of the adenosine bases in which the target protein is Hsp70. As with several previously reported base modifications, the position of the modification is away from the hydrogen bonding face of the adenosine base and therefore does not interfere with base-pairing, making it suitable for use in SELEX with polymerases like KOD XL that can accept such modifications. Synthesis of the modified base is simple and elegant, with both nucleoside triphosphates (needed for SELEX) and phosphoramidites (needed for chemical synthesis) reported in sufficient detail. The selection method, binding properties, and specificity of the resulting modified aptamers is presented in a systematic and clear way, including the testing of appropriate control unmodified aptamers with the same sequence but lacking the modified side chains. The authors demonstrate that the modifications are essential for high-affinity binding and therefore add another type of a diversity-enhancing option to the armamentarium of base modifications available for SELEX, with the goal of being able to identify aptamers with improved binding properties. As such, the paper will be of interest to a wide audience of readers of Communications Chemistry. The authors should consider the following points:

Major point:

- The sequences in the evolved region shown in Table 3 should be aligned to show conserved motifs, which clearly exist. Without alignment, this is very difficult to follow. The use of an asterisk to indicate the modification (which is otherwise fine throughout the manuscript) complicates the alignment, but this can be fixed in a number of ways, including by using an alternative single letter just for this purpose. Easy inspection of the conserved and variable positions could indicate the importance of all evolved position for binding.

Minor points:

- An intermediate compound referred to as dAEEPh on the path toward the final product (referred to as dA* for convenience) has an unsaturation (alkynyl moiety) adjacent to the 7-position of adenosine (Scheme 1 on page 6). This introduces a restriction in rotational degrees of freedom at that position

that pushes the side chain away from the base in a manner that makes it an interesting modification in its own right. The authors may wish to comment on why that modification was not considered as a separate type of a modification, or why the fully saturated alternative was preferred.

- On line 136, page 7, the authors mention the “random 65nt library.” This can be interpreted as referring to a random library that has 65 randomized positions, whereas the number of randomized positions is 28 (as indicated in Table 1). This should be clarified.
- In Figure 2, dA*TP as well as dABuPh are used to describe the same modification. Only one should be used (probably dA*TP).
- Selection conditions are nicely summarized in Table 2. This includes the use of salmon sperm DNA in rounds 4 through 9. However, the amount of salmon sperm DNA is not indicated in either the text of the paper or the experimental section. Since salmon sperm DNA presumably serves as a polyanionic nonspecific competitor that increases the stringency of selections, some experimental detail should be provided.
- The authors state in the Discussion section on lines 291-294 that the “reason for the enhancement could be due to a change in the folded conformation...[or] due to some direct interactions of the hydrophobic moieties with the protein.” This is an important point, for which there is now considerable experimental support from co-crystal structures of base-modified aptamers with several target proteins. This should be at least mentioned in the text.
- Binding of aptamers HSc-2 and HSc-9 is assessed by two independent methods which show somewhat different values. This in itself is not a concern. However, indicating such differences in nM units (for example, “a difference of almost 200 nM” on line 301) is unusual and should be changed to, for example, fold-difference between the observed Kd values.
- The first sentence of the Conclusion section on lines 397-398, which indicates the importance of hydrophobic modified aptamers, points to reference 24. This outstanding paper, Li et al. 2008, reports the use of boronic acid modifications as a way of targeting carbohydrate moieties on glycoproteins (through covalent interactions), so it may not be the best example for the intended point. The authors may wish to include several other papers already cited in the paper as examples of the importance of hydrophobic modifications that facilitate binding through non-covalent interactions.

Response to reviewers and list of changes manuscript COMMSCHEM-22-0555

Reviewers' comments:

Reviewer #1 (Remarks to the Author):

The study by Hocek and co-workers describes the identification of DNA aptamers with hydrophobically modified adenine residues binding to Hsp70. The study describes the synthesis of the modified ATP and phosphoramidite, the use of these to generate modified DNA libraries and their application to SELEX for enriching the respective aptamers. The enriched libraries were analysed regarding sequence content and the interaction properties of one representative aptamer are shown in the main part of the manuscript, including truncation and proof of interaction-dependency on the hydrophobic modification. The study is very well written and described in a scholar manner, describes a modified aptamer binding to Hsp70 and might be acceptable for publication, provided the comments below are considered:

1. references of relevant studies in the fields are given but this reviewer missed the work by Hirao on hydrophobic base-pairs and Mayer on click-modified (hydrophobic) modified aptamers/clickmers, which have proven useful as potential therapeutics and diagnostics.

Revision: References added:

(24) Hirao, I. Kimoto, M. Mitsui, T. Fujiwara, T. Kawai, R. Sato, A. Harada, Y. Yokoyama, S. An unnatural hydrophobic base pair system: site-specific incorporation of nucleotide analogs into DNA and RNA. *Nature Methods*. **3**, 729-735 (2006)

(25) Pfeiffer, F., Tolle, F., Rosenthal, M., Brändle, G.M., Ewers, J., Mayer, G. Identification, and characterization of nucleobase modified aptamers by click- SELEX. *Nature Protocols*. **13**, 1153-1180 (2018)

2. Abstract: Typo Aptamers needs to be aptamers

Revision: Typo corrected

3. Controls: the study shows data using a naive DNA aptamer (which revealed fairly high affinity to Hsp70) and the corresponding (selected) modified aptamer with much higher affinity. However, missing here data on scrambled versions of the modified aptamer, having the same amount of modifications and of each other nt, but in a different order. The Aptamer Society recently published a white paper on minimal criteria necessary to meet prior to publish new aptamer/target pairs, and such controls are identified as essential.

Revision: Three scrambled sequences of aptamer HSc-9 were synthesized and tested by BLI and fluorescent plate assay. Their K_d values are displayed in Table 5 and are at least one order of magnitude higher than of the HSc-9. This clearly confirms the fact that the sequence (not only the presence of modifications) is needed for the high affinity of the aptamer.

4. Along the same lines as 3: Which modifications at which positions are important for maintaining binding? Are all 13 modifications necessary or can be some of them omitted? This would be valuable data for further studies on the aptamer but might go beyond the scope of this ms.

Response: we agree that this would certainly be valuable information on the aptamer, but it is out of the scope of this manuscript. There are too many combinations that would need to be synthesized and tested to verify this.

Reviewer #2 (Remarks to the Author):

In this manuscript the authors present the selection of hydrophobic modified DNA aptamers against HSP70. The paper incorporates rigorous chemistry of the synthesis of a modified dATP which has a hydrophobic sidechain. This was used in a SELEX to discover hydrophobic modified aptamers, then aptamers characterised by various biophysical methods. The chemistry of the paper is very strong, overall I would consider this could be suitable for Comms Chem - ideally this might include a structure of the complex as well, but the extent of characterisation is also very good, although not quite exceptional without X-ray structure.

I have these particular comments to address:

1. During selection it seems only magnetic particles instead of a different protein+magnetic particle was used as counterselection. Therefore, one must be careful with controls that this is not a general protein binder, particularly when considering the hydrophobic modification.

Response: Counter selection was carried out with magnetic particles and an alternative protein (His-tagged NPA). Respectfully, the reviewer may have missed this information. This is highlighted in the Table 2 and is explained by “[b] Negative selection with His-NPA before incubation with the target”. Moreover, we tested the aptamer HSc-9.1 against three alternative His-tagged proteins where it showed $K_d > \mu\text{M}$ which excludes the possibility of being a general protein binder.

2. BLI data (eg. S13) should be shown with an alternate protein to HSP70 as a control. In both directions (ie. also S14). Also figures 4 and 5 - these need more controls of alternative proteins and alternative DNA sequences overlaid.

Revision: we performed additional studies with alternative His-tagged proteins, an SBP-tagged Hsp70 and with 3 scrambled sequences (see response to Reviewer 1). These additional studies

confirmed both target and sequence specificity of our aptamer.

2. I would advise switching some data of binding to main figures. The raw data of S13 and S15 should be the main figure with the graph prism export figure 4 and 5 being sub-figures within that to allow the reader to see quality of raw data without referring supplementary.

Response: we prefer to have the detailed raw data in Supplementary Information. In the binding curves of Figures 5 and 6 we show SD so the reader can see the quality of the data.

3. Specificity is shown in Figure 6 but this is more qualitative at a single concentration instead of quantitative. Ideally this should actually provide a K_d to each of the targets (or none shown if cannot be determined) through a range of concentrations.

Revision: Further studies were performed on alternative His-tagged proteins and the K_d values are now determined and displayed in Table 4.

4. Is there data the aptamer can bind when the his tag is removed from Hsp70?

Revision: We added a BLI experiment with HSc-9.1 and SBP-tagged Hsp70 and was found to bind with similar affinity to the His-tagged Hsp70. (Figure S21 and Table 4).

5. A table showing relative affinities of different sequences with and without modification would be helpful - it is difficult to dig this out of the text describing Fig 4 and Fig 5 on page 15.

Also the comparison of BLI and fluorescent plate data.

Revision: All determined binding affinities are now summarized in Tables 4 and 5 for easy comparison.

6. For Table 3 is there some sort of alignment? Are these sequences related?

Revision: We added Figure 4 with alignment of the sequences.

7. A little more discussion regarding biological relevance of aptamers to detect Hsp70 would be helpful.

Revision: Further discussion of this was added in the introduction section along with references 32-35.

In summary this is quite a strong paper particularly from the chemistry perspective. In my view after revisions, it may be suitable for publication.

Reviewer #3 (Remarks to the Author):

The use of nucleic acid libraries with modifications that increase chemical diversity has

substantially broadened the scope of targets for which high-affinity aptamers can be identified. Mulholland et al. report the selection of DNA-based aptamers with a new hydrophobic aromatic modification introduced at the 7-position of the adenosine bases in which the target protein is Hsp70. As with several previously reported base modifications, the position of the modification is away from the hydrogen bonding face of the adenosine base and therefore does not interfere with base-pairing, making it suitable for use in SELEX with polymerases like KOD XL that can accept such modifications. Synthesis of the modified base is simple and elegant, with both nucleoside triphosphates (needed for SELEX) and phosphoramidites (needed for chemical synthesis) reported in sufficient detail. The selection method, binding properties, and specificity of the resulting modified aptamers is presented in a systematic and clear way, including the testing of appropriate control unmodified aptamers with the same sequence but lacking the modified side chains. The authors demonstrate that the modifications are essential for high-affinity binding and therefore add another type of a diversity-enhancing option to the armamentarium of base modifications available for SELEX, with the goal of being able to identify aptamers with improved binding properties. As such, the paper will be of interest to a wide audience of readers of Communications Chemistry. The authors should consider the following points.

Major point:

- The sequences in the evolved region shown in Table 3 should be aligned to show conserved motifs, which clearly exist. Without alignment, this is very difficult to follow. The use of an asterisk to indicate the modification (which is otherwise fine throughout the manuscript) complicates the alignment, but this can be fixed in a number of ways, including by using an alternative single letter just for this purpose. Easy inspection of the conserved and variable positions could indicate the importance of all evolved position for binding.

Revision: The alignment of all the full length sequences is now shown in Figure 4 and common motif has been properly indicated for better visualization.

Minor points:

An intermediate compound referred to as dAEEPh on the path toward the final product (referred to as dA* for convenience) has an unsaturation (alkynyl moiety) adjacent to the 7-position of adenosine (Scheme 1 on page 6). This introduces a restriction in rotational degrees of freedom at that position that pushes the side chain away from the base in a manner that makes it an interesting modification in its own right. The authors may wish to comment on why that modification was not considered as a separate type of a modification, or why the fully saturated alternative was preferred.

Revision: We added a comment on the design of the modification:

The 7-phenylbutyl-7-deaza-2'-deoxyadenosine was designed as a suitable modification of aptamers because the phenylbutyl moiety contains an aromatic ring and flexible and hydrophobic tether that might allow hydrophobic, π - π stacking or cation- π interactions with the target protein.

Otherwise we agree that the alkyne-linked modification is certainly also interesting and may be used in other selection studies in the future.

- On line 136, page 7, the authors mention the “random 65nt library.” This can be interpreted as referring to a random library that has 65 randomized positions, whereas the number of randomized positions is 28 (as indicated in Table 1). This should be clarified.

Revision: The sentence has been re-structured and clarified.

- In Figure 2, dA*TP as well as dABuPh are used to describe the same modification. Only one should be used (probably dA*TP).

Revision: The figure has been corrected.

- Selection conditions are nicely summarized in Table 2. This includes the use of salmon sperm DNA in rounds 4 through 9. However, the amount of salmon sperm DNA is not indicated in either the text of the paper or the experimental section. Since salmon sperm DNA presumably serves as a polyanionic nonspecific competitor that increases the stringency of selections, some experimental detail should be provided.

Revision: Detailed information was added to Table 2 and selection procedure in Materials and methods.

- The authors state in the Discussion section on lines 291-294 that the “reason for the enhancement could be due to a change in the folded conformation...[or] due to some direct interactions of the hydrophobic moieties with the protein.” This is an important point, for which there is now considerable experimental support from co-crystal structures of base-modified aptamers with several target proteins. This should be at least mentioned in the text. Binding of aptamers HSc-2 and HSc-9 is assessed by two independent methods which show somewhat different values. This in itself is not a concern. However, indicating such differences in nM units (for example, “a difference of almost 200 nM” on line 301) is unusual and should be changed to, for example, fold-difference between the observed K_d values.

Revision: This sentence has been modified.

- The first sentence of the Conclusion section on lines 397-398, which indicates the importance of hydrophobic modified aptamers, points to reference 24. This outstanding paper, Li et al. 2008, reports the use of boronic acid modifications as a way of targeting carbohydrate moieties on glycoproteins (through covalent interactions), so it may not be the best example for the intended point. The authors may wish to include several other papers already cited in the paper as

examples of the importance of hydrophobic modifications that facilitate binding through non-covalent interactions.

Revision: References have been changed to reference 7, 21 and 24 which support the point better.

REVIEWERS' COMMENTS:

Reviewer #1 (Remarks to the Author):

I read the revised version of the manuscript and find significantly improved, addressing all criticism raised by the reviewers and, thus, would suggest acceptance for publication as is.

Reviewer #2 (Remarks to the Author):

All comments have now been addressed in the revised submission.

Reviewer #3 (Remarks to the Author):

Thank you for addressing all of my suggestions and comments.